# Single-cell RNA-sequencing reveals pre-meiotic X-chromosome dosage compensation in *Drosophila* testis

**Evan Witt**[1], **Zhantao Shao**[2], **Chun Hu**[2], **Henry M. Krause**[2]*, **Li Zhao**[1]*

**1** Laboratory of Evolutionary Genetics and Genomics, The Rockefeller University, New York, New York, United States of America, **2** Department of Molecular Genetics and The Donnelly Centre for Cellular and Biomolecular Research, University of Toronto, Toronto, Ontario, Canada

* h.krause@utoronto.ca (HMK); lzhao@rockefeller.edu (LZ)

**Data Availability Statement:** R Code for integrating the Seurat objects for the 517 and wild strains, the mathematical analyses, and generation of figures is available at https://github.com/LiZhaoLab/Singlecell_DosageCompensation. Fastq

## Abstract

Dosage compensation equalizes X-linked expression between XY males and XX females. In male fruit flies, expression levels of the X-chromosome are increased approximately two-fold to compensate for their single X chromosome. In testis, dosage compensation is thought to cease during meiosis; however, the timing and degree of the resulting transcriptional suppression is difficult to separate from global meiotic downregulation of each chromosome. To address this, we analyzed testis single-cell RNA-sequencing (scRNA-seq) data from two *Drosophila melanogaster* strains. We found evidence that the X chromosome is equally transcriptionally active as autosomes in somatic and pre-meiotic cells, and less transcriptionally active than autosomes in meiotic and post-meiotic cells. In cells experiencing dosage compensation, close proximity to MSL (male-specific lethal) chromatin entry sites (CES) correlates with increased X chromosome transcription. We found low or undetectable levels of germline expression of most *msl* genes, *mle*, *roX1* and *roX2* via scRNA-seq and RNA-FISH, and no evidence of germline nuclear *roX1/2* localization. Our results suggest that, although dosage compensation occurs in somatic and pre-meiotic germ cells in *Drosophila* testis, there might be non-canonical factors involved in the dosage compensation mechanism. The single-cell expression patterns and enrichment statistics of detected genes can be explored interactively in our database: https://zhao.labapps.rockefeller.edu/gene-expr/.

## Author summary

Male flies need to boost gene expression from their single X chromosome to equal that of females, which have two X chromosomes. In this process, called dosage compensation, the dosage compensation complex binds to genomic chromatin entry sites and upregulates gene expression nearby. This process was thought to be restricted to somatic cells. Using single-cell RNA-seq data, we found that certain germ cell types in the *Drosophila* testis show X chromosome expression similar to that of the autosomes, implying dosage compensation activity. In these cell types, we found evidence that genes near a chromatin

files of the single-cell testis RNA-seq data have been deposited at NCBI SRA with accession numbers SAMN10840721 (RAL517 strain, BioProject # PRJNA517685) and SAMN12046583 (Wild strain, PRJNA548742). The single-cell-level expression pattern and statistics of each gene can be found in the database: https://zhao.labapps. rockefeller.edu/gene-expr/.

**Funding:** Funding for work performed in the lab of L.Z. was provided by NIH MIRA R35GM133780, the Robertson Foundation, a Monique Weill-Caulier Career Scientist Award, an Alfred P. Sloan Research Fellowship (FG-2018-10627), a Rita Allen Foundation Scholar Program, and a Vallee Scholar Program (VS-2020-35) to L.Z.; Funding for work performed in the lab of H.M.K. was provided by the Canadian Institutes of Health Research (PJT-165884) to HMK. The funders had no role in study design, data collection and analysis, decision to publish, or preparation of the manuscript.

**Competing interests:** The authors have declared that no competing interests exist.

entry site are more highly expressed than genes farther away, which is additional evidence of dosage compensation. In cell types without evidence of dosage compensation, we saw no evidence of chromatin entry site activity. Interestingly, we found little evidence of expression of most genes from the dosage compensation complex using both RNA-FISH and scRNA-seq. This suggests that our observed pre-meiotic dosage compensation is likely to be mediated by a noncanonical mechanism. These findings add new insight into our understanding of sex chromosomes.

## Introduction

In a wide variety of sexually reproducing animals, males and females have different numbers of X chromosomes. Somatic expression of X-linked genes needs to be adjusted so that males and females produces similar levels of most proteins encoded on the variable chromosome [1]. Indeed, for many sexually reproducing animals, transcription is adjusted to compensate for differing numbers of sex chromosomes, a phenomenon referred to as dosage compensation [2,3]. Strategies for this process vary dramatically across the animal kingdom, some to increase chromosome X transcription in males, others to randomly inactivate one X or to partially suppress both X chromosomes in females [4–8]. The spatial and temporal patterns of dosage compensation are key to understanding gene expression regulation and its roles in development. ScRNA-seq has been successfully used to study dosage compensation during development [9], but has not yet been applied to adult spermatogenesis.

In somatic cells from male *Drosophila*, the dosage compensation complex (DCC) (including the Male Sex Lethal proteins MSL-1, MSL-2 MSL-3, Maleless (MLE), and males absent on the first (MOF)) acts in concert with Chromatin-linked adaptor for MSL proteins (CLAMP) [10] and two noncoding RNAs, *roX1* and *roX2*, to increase transcription from the male X chromosome to levels comparable to the paired autosomes [4,8,11]. This binding is facilitated by the MSL proteins, which bind to a specific sequence motif in chromatin entry sites (CES) on the X chromosome and spread outward, upregulating local genes [12,13]. During male meiosis in some animal species, the sex chromosomes are downregulated in excess of that expected from the loss of dosage compensation. This is referred to as Meiotic Sex Chromosome Inactivation (MSCI) [14]. Whereas the absence of dosage compensation would cause a 50 percent drop in X transcription, MSCI actively represses the X even further, leading to less than 50 percent X activity.

Dosage compensation in *Drosophila* somatic tissues (such as the brain) has been extensively studied [11,13,15]. However, whether dosage compensation occurs in germ cells (testis) and to what extent it plays a role in spermatogenesis, is actively debated. For instance, prior work has raised the possibility of germline dosage compensation [16,17], although the magnitude and timing of this process are unclear. Other work found that male-biased X chromosome genes are usually found outside of dosage compensated regions, suggesting that dosage compensation does influence patterns of sex-biased expression on the X chromosome [18]. Further work suggests that both dosage compensation and MSCI are absent during *Drosophila* spermatogenesis [19]. Unlike in somatic cells, the MLE protein does not associate specifically with the X chromosome in male germ cells, suggesting a possible alternative function of the *mle* gene in germ cells [20]. Given that the MSL complex is thought not to localize to male germline X chromosomes [20], the mechanism of germline dosage compensation in *Drosophila* male germ cells remains a mystery [9].

MSCI and dosage compensation are difficult to study in *Drosophila* because both somatic and sex chromosomes are downregulated during and after meiosis, making it challenging to identify groups of genes with deviant expression patterns in the context of global transcriptional downregulation. In some cases it may be difficult to distinguish the effect of X chromosome inactivation from the loss of dosage compensation. However, multiple transgenic insertions in X and autosomes [17,21] show that X inactivation exceeds that expected for loss of canonical dosage compensation. The researchers found no evidence that any region of the X chromosome escapes dosage compensation. Another group found X suppression of 2–4 fold in the male germline, ruling out loss of dosage compensation as the sole cause of X inactivation [22]. New evidence supports spermatogonial dosage compensation in scRNA-seq from larval testis [9]. We asked if the same is true in adult testis by analyzing gene expression from somatic, pre-meiotic, meiotic, and post-meiotic cells. We also asked if there was evidence of *roX1/2* activity in germline cells experiencing dosage compensation.

We sought to quantify the relative transcriptional dynamics of sex chromosomes and autosomes in a high throughput fashion with scRNA-seq of testis from two strains of *Drosophila melanogaster*. Compared to dissection-based methods, our methods allow high confidence in the identities of assigned germline cell types from somatic cell types, allowing us to holistically investigate germline X and autosome activity with greater precision. We found that pre-meiotic cells and somatic cells show X:autosome expression ratios close to 1, consistent with dosage compensation in pre-meiotic cells. After meiosis, X:autosome ratios decline to around 0.6–0.7, indicative of incomplete dosage compensation. Our observed pre-meiotic dosage compensation occurs despite negligible germline expression of *roX1* and *roX2*, which we confirmed with RNA-FISH. We also found that, in somatic and early germ cell types that have balanced X/autosome output, genes within 10,000 bp of an MSL CES have higher expression than genes further away. Our results support the existence of pre-meiotic X-chromosome dosage compensation in *Drosophila* testis via a noncanonical mechanism.

## Results

### Total transcription of X and autosomes peaks in early germ cells and is reduced after meiosis

We analyzed two single-cell testis RNA-seq datasets that we generated [23], totaling 13,000 cells. We chose to use two different strains to ensure that our findings are robust with respect to technical and strain variation, and repeated the analyses of each strain separately to ensure that all findings were reproducible (see Methods). Using the ScTransform function from Seurat V3 [24], we combined data from the two *D. melanogaster* strains and clustered corresponding cell types together on the same set of axes (Fig 1A and 1B). Consistent with our previous report, the two datasets correlate with a Pearson's R of 0.97, and cells from both strains overlap well, with a qualitatively good distribution of cells from both strains around our dimensionally reduced dataset used for clustering [23] (S1 Fig). Compared to our previous report, in which we only used one strain of *D. melanogaster*, we were able to improve upon our previous cell type assignments by using a more accurate marker gene to denote cyst cells, *Rab11* [25] (S2 and S3 Figs). Asking if the X and autosomes both follow expected trends of post-meiotic transcriptional downregulation, we counted the Unique Molecular Indices (UMIs, a proxy for RNA content) from X and autosomes in all cells. We observed that total RNA per cell from the X and autosomes peaks in late spermatogonia and early spermatocytes and is then reduced in late spermatogenesis (Figs 1C and 1D, S4 Fig). If a cell type were dosage compensated, we would expect this ratio of X/A (X/autosome) to be similar to or higher than that found in somatic cells. If a cell type lacked dosage compensation, we

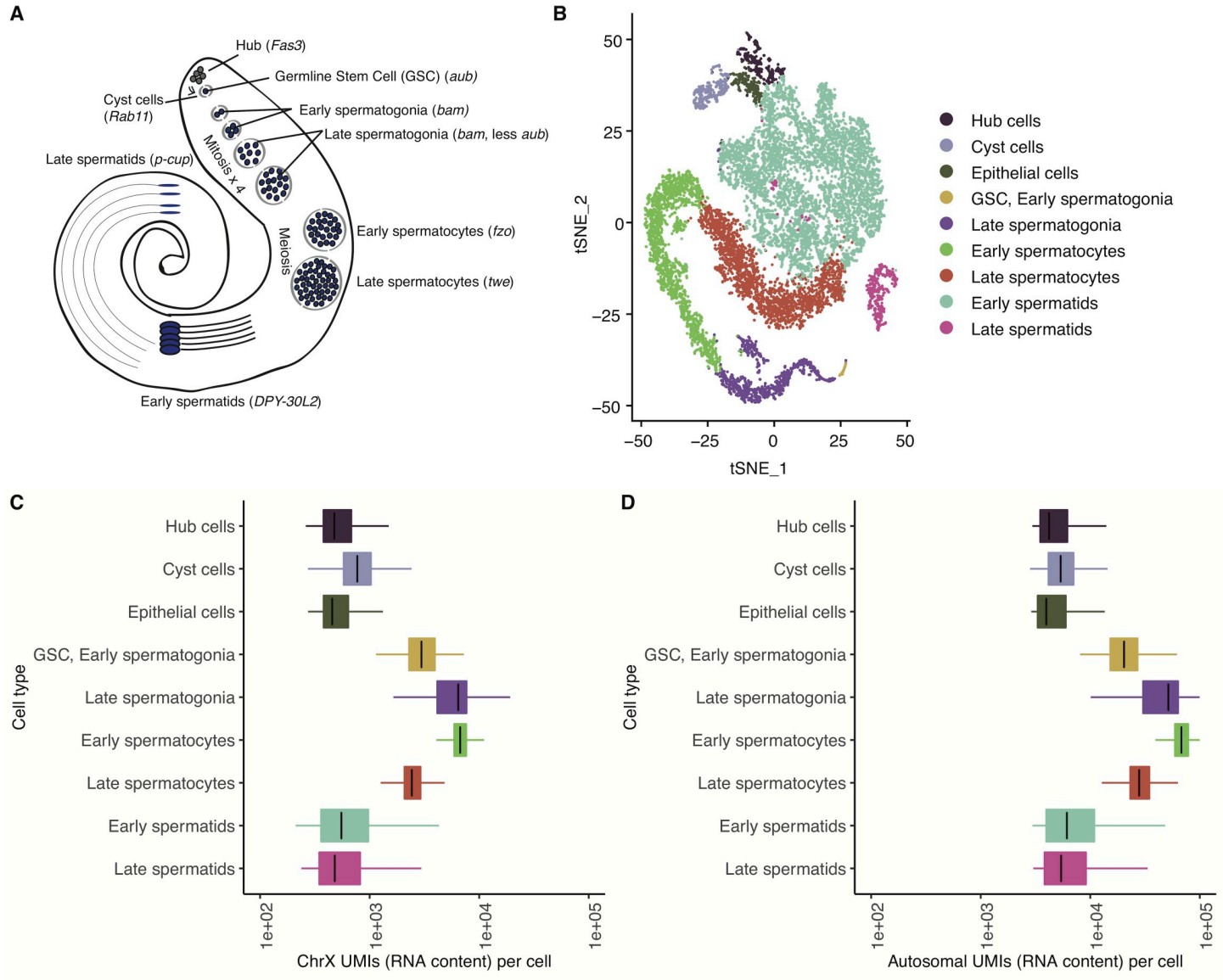

**Fig 1. Integrated analysis of testis sequencing from two *D. melanogaster* strains.** A) A schematic showing the types of cells present inside *Drosophila* testis. Germline stem cells differentiate into spermatogonia, which undergo mitosis and become spermatocytes, which undergo meiosis and become haploid spermatids. B) t-SNE plot showing a dimensional reduction of our combined, normalized libraries. Each point is a cell, clustered according to its similarity to other cells. C and D) For every cell, the number of RNA molecules detected from the X chromosome and autosomes approximated by the summed number of Unique Molecular Indices counted per cell. X and autosomes are both downregulated during late spermatogenesis. Count numbers and ratios are shown in S1 Table.

would expect this ratio to be lower than that of somatic cells. Taking a closer look, although the overall patterns for X and autosome look very similar (Fig 1C and 1D), the ratios of X/autosome are the highest in GSC and spermatogonia (0.120–0.145), intermediate in somatic hub, cyst, and epithelial cells (0.104–0.135), and the lowest in meiotic (spermatocytes) and post-meiotic (spermatids) cells (0.087–0.090) (S1 Table). The relatively reduced ratio of X chromosome RNA in spermatocytes and spermatids suggests that the X chromosome is downregulated in excess of autosomes in meiotic and post-meiotic cells. The observation that this X/autosome ratio is higher in early germ cells than somatic cells indicates some form of excess dosage compensation.

### Relative RNA production from the X chromosome indicates dosage compensation in pre-meiotic cells

Since the total RNA content of a cell is an imperfect measure of gene expression trends, we asked whether the RNA content produced by individual genes from the X chromosome and autosomes is roughly equivalent in different stages of spermatogenesis. To ask whether a stage favors X or autosomes, we counted the detected reads per gene in each cell type for X and autosomes. If a cell type produces roughly equal amounts of RNA from X and autosomal genes, we would consider that cell type subject to dosage compensation. If a cell type produced less RNA from a median X gene than an autosomal gene, we would interpret this as the absence of dosage compensation, similar to earlier work [21].

In meiotic (spermatocyte) and post-meiotic (spermatid) cells, we observe a depletion in the relative RNA produced from the X chromosome. We found that the median reads per gene were lower for the X than autosomes in early and late spermatocytes, and early and late spermatids in both fly strains (Figs 2 and S5, Table 1, Holm-adjusted p values: 4.82e-07, 5.35e-07,1.16e-04 and 6.50e-04, respectively). Interestingly, X and autosomes produced roughly equal numbers of reads per gene in somatic hub, cyst, and epithelial cells, and germline late spermatogonia (adjusted p values: 2.86e-01, 8.24e-02, 2.86e-01, 8.24e-02, respectively). In GSC

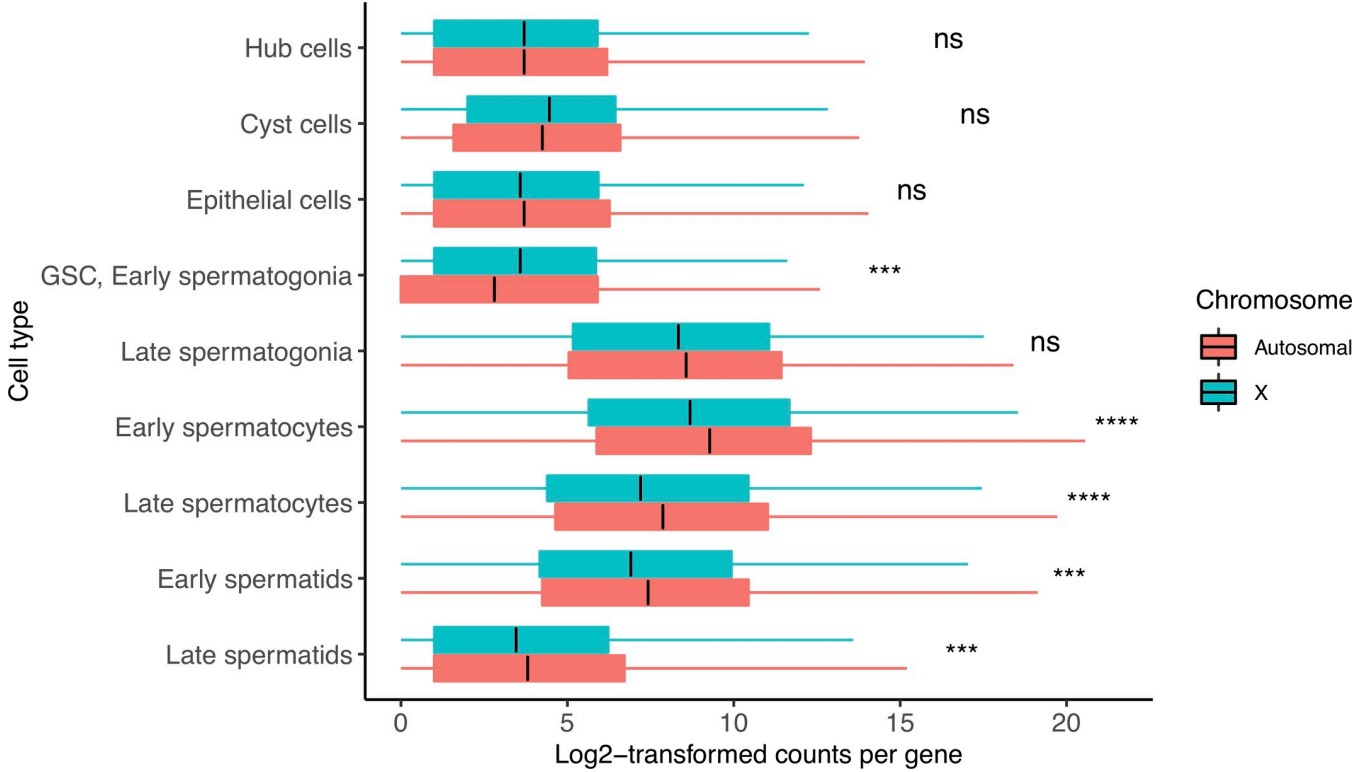

**Fig 2. Dosage compensation equalizes X and autosomal transcription in select cell types.** Shown is the median log-transformed (log2(counts+1) counts for every gene in X and autosomes. This ratio is similar in hub, cyst, epithelial cells, and late spermatogonia, indicating the presence of dosage compensation. GSC and early spermatogonia show a statistical enrichment of X chromosome transcription, indicating excess dosage compensation. Spermatocytes and spermatids have fewer X chromosomal than autosomal transcripts per gene, indicating that dosage compensation has been lost. This loss coincides with reduced expression of dosage compensation genes as seen in Fig 4A. Asterisks represent Holm-adjusted p values of a two-way Wilcoxon test of the null hypothesis that autosomal genes and X genes have equal detectable log-normalized counts in a cell type. In GSC, Early spermatogonia, we see a significant likelihood that the means of the two groups are not equal, and we can see that it is due to an enrichment of X chromosome counts. In meiotic cells this is the opposite- the significant value represents a depletion of X counts, since this is not a directional Wilcoxon test. Asterisks represent p values as follows: ns: >0.05, *<0.05, **<0.005, ***<0.0005, ****<0.00005. Detailed raw and adjusted p values, and interpretations of results of each cell type are in Table 1.

**Table 1. X and autosomal count ratios detected per gene for every cell type.** These ratios are of raw counts from X and autosomes, not log transformed. The ratio of X/autosome expression is close to or greater than 1 in hub, cyst, GSC, early spermatogonia and late spermatogonia, indicating the presence of dosage compensation. Late spermatogonia may also be interpreted as no DC if they contain most transcripts generated from early spermatogonia. This ratio decreases in meiotic and post-meiotic cells, indicating the loss of dosage compensation. P values are for a two-sided Wilcoxon test with a null hypothesis that X and autosome genes are similarly expressed, interpreted as dosage compensation. Meiotic and post-meiotic cells show an imbalance in x-autosome ratios, indicative of absent dosage compensation, but in pre-meiotic and somatic cells, x and autosomal ratios are near or over 1. The X/autosome ratio is highest in GSC/early spermatogonia, with a statistical enrichment of X chromosome gene expression, interpreted as possible excess dosage compensation in this cell type. DC means dosage compensation.

| Cell type | X/autosome ratio of raw counts | Raw p value | Adjusted P value | Interpretation |
|---|---|---|---|---|
| Hub cells | 1.00 | 1.43e-01 | 2.86e-01 | DC |
| Cyst cells | 1.17 | 2.34e-02 | 8.24e-02 | DC |
| Epithelial cells | 0.92 | 1.63e-01 | 2.86e-01 | DC |
| GSC, Early spermatogonia | 1.83 | 6.44e-05 | 3.86e-04 | Excess DC |
| Late spermatogonia | 0.85 | 2.06e-02 | 8.24e-02 | DC |
| Early spermatocytes | 0.66 | 5.35e-08 | 4.82e-07 | No DC |
| Late spermatocytes | 0.63 | 6.69e-08 | 5.35e-07 | No DC |
| Early spermatids | 0.70 | 1.65e-05 | 1.16e-04 | No DC |
| Late spermatids | 0.77 | 1.30e-04 | 6.50e-04 | No DC |

and early spermatogonia, the X:autosome ratio increases to 1.83, suggestive of a possibility of excess dosage compensation (adjusted p value: 3.86e-04), although the pattern of excess dosage compensation might partly be caused by up-regulation of X-linked testis-specific genes (see below). Alternatively, it might be partly caused by global chromatin remodeling of the X during these stages. The apparent dosage compensation in late spermatogonia could also be caused by leftover transcripts from the surge of X chromosome RNA found in GSC/early spermatogonia. We repeated this analysis with gene expression values linearly scaled from 0 to 1 and found similar results (S6 Fig).

## Gene expression is enriched for genes close to a CES

The DCC is thought to facilitate dosage compensation by binding to specific DNA motifs and spreading outward, upregulating local genes [13]. We examined transcriptional activity for every gene approximated by the total counts detected in a cell type and found evidence that, in certain cell types, genes close to a CES have higher RNA output (counts) than genes further away (Figs 3, S7, and S8, Tables 2 and S2). These cell types are hub, cyst, epithelial, GSC, early spermatogonia, and late spermatogonia, (Adjusted p values for two-tailed Wilcoxon test = 7.42e-04, 1.24e-08, 1.59e-03, 4.06e-08, and 3.82e-02, respectively) the same cell types for which we observed evidence of dosage compensation in Fig 2. In cell types without evidence of DC, there is not a significant difference in detectable counts per gene between genes <10000 bp from a CES, and genes further away. The same trend is present in somatic cells for which we observed dosage compensation, and also in pre-meiotic germ cells. Pearson's correlations between distance and expressed counts are higher for somatic and pre-meiotic cells than meiotic and post meiotic cells (S9 Fig). This suggests that our observed dosage compensation in somatic and pre-meiotic cells is mediated from the same sequence element, with possible different mechanisms.

## Testis-biased and testis-specific genes do not bias apparent dosage compensation

To understand whether testis-biased genes influence the observed patterns of dosage compensation [26], we repeated the analyses from Figs 1, 2 and 3 after removing testis-specific or testis-biased genes from our data. S8, S10 and S11 Figs show that the appearance of dosage

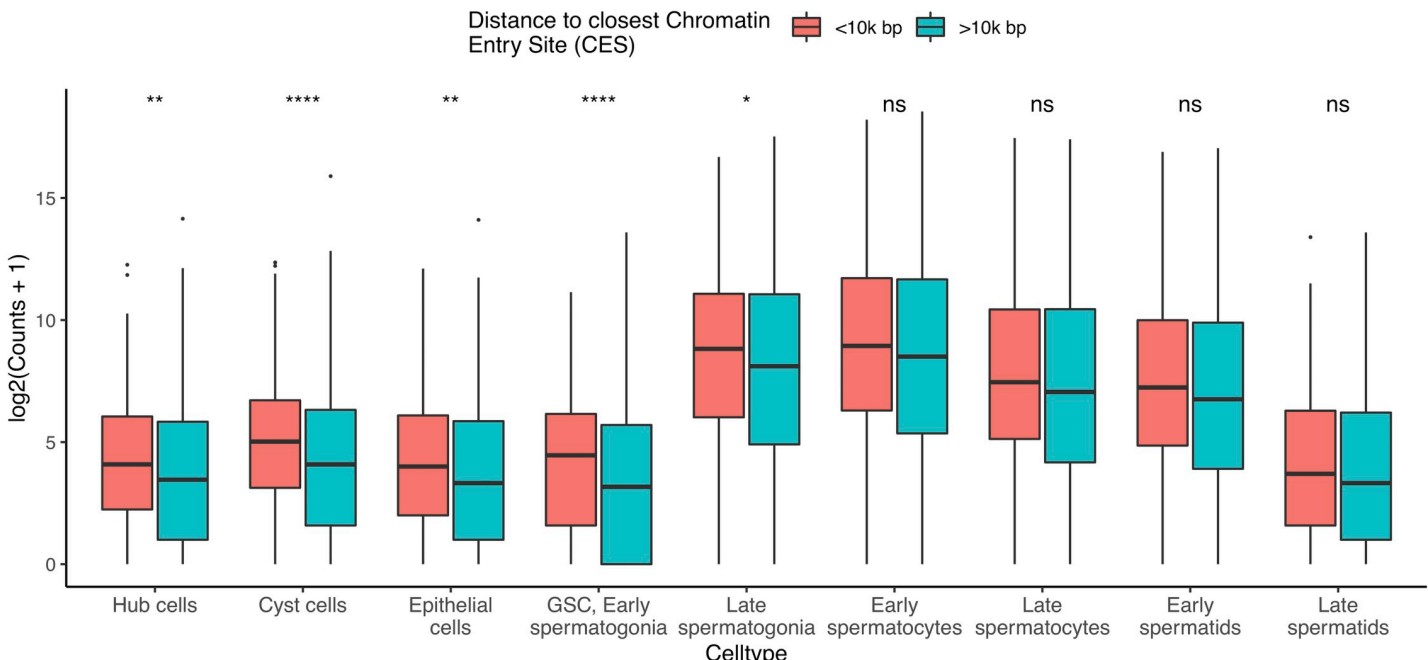

**Fig 3. Close MSL CES correlates with increased transcription of X chromosome genes in cell types experiencing dosage compensation.** The Y axis is log-transformed sum of counts for a given X chromosome gene in a cell type, grouped by proximity to a CES. Genes within 10,000 bp of a CES have statistically enriched expression compared to genes outside this range in all somatic cell types, and pre-meiotic germ cells (p values for two-tailed Wilcoxon test in Table 2). These are the same cell types with statistically similar X and autosomal expression in Fig 2. This indicates that the DCC is active in these cell types. Asterisks represent Holm-corrected p values as follows: ns: >0.05, * <0.05, ** <0.005, *** <0.0005, **** <0.00005.

compensation in somatic and pre-meiotic cells remains without these genes. After removing testis-specific or testis-biased genes in the analysis, genes near CES still show a higher level of expression in GSC and early spermatogonia (S8 Fig). One difference is that when testis-specific genes are removed from the analysis, there is no longer evidence of excess dosage compensation in GSC/early spermatogonia (S11 Fig), indicating that these genes are disproportionately upregulated in these cells. Without these testis-specific genes, the distributions of X and autosome counts are similar, consistent with the degree of dosage compensation observed in

**Table 2. Evidence of DCC activity in somatic cells and pre-meiotic germ cells.** In each cell type we have calculated the median counts for X chromosome genes within 10000 bp of an annotated CES and genes outside 10000 bp, corresponding to Fig 4. In hub, cyst, epithelial, GSC, early and late spermatogonial cells, a two-tailed Wilcoxon test finds that these two sets of genes have statistically different count distributions with an alpha of 0.05. These cell types have a higher ratio of median counts from close and distant genes, indicating that CES sites influence X chromosome gene expression in these cell types, a hallmark of dosage compensation. The cell type with the highest difference between close/distant x gene expression is GSC/early spermatogonia, which also had the highest X/autosome ratio in Table 1. DCC means dosage compensation complex.

| Cell type | <10k bp, median counts | >10k bp, median counts | Close/ distant median ratio | Raw p value | Adjusted p value | Interpretation |
|---|---|---|---|---|---|---|
| **Hub cells** | 16 | 10 | 1.6 | 1.06E-04 | 7.42E-04 | Active DCC |
| **Cyst cells** | 31.5 | 16 | 1.97 | 1.38E-09 | 1.24E-08 | Active DCC |
| **Epithelial cells** | 15 | 9 | 1.67 | 2.65E-04 | 1.59E-03 | Active DCC |
| **GSC, Early spermatogonia** | 21 | 8 | 2.63 | 5.08E-09 | 4.06E-08 | Active DCC |
| **Late spermatogonia** | 451.5 | 275 | 1.64 | 7.65E-03 | 3.82E-02 | Active DCC |
| **Early spermatocytes** | 490.5 | 362 | 1.35 | 7.69E-02 | 1.54E-01 | Inactive DCC |
| **Late spermatocytes** | 174.5 | 132 | 1.32 | 8.41E-02 | 1.54E-01 | Inactive DCC |
| **Early spermatids** | 150 | 107 | 1.4 | 3.99E-02 | 1.20E-01 | Inactive DCC |
| **Late spermatids** | 12 | 9 | 1.33 | 2.31E-02 | 9.24E-02 | Inactive DCC |

somatic cells. This could indicate an uneven degree of dosage compensation for different gene classes in early germ cells. Whether the excessive up-regulation of testis-biased genes is through dosage compensation or other mechanism is unclear. However, it will be important to study the mechanism and the impact of this pattern in the future.

### Germline dosage compensation takes place despite low levels of *roX1* and *roX2*

We sought to explain our observed testis somatic and pre-meiotic dosage compensation by querying the expression patterns of genes known to regulate this process in somatic cells. The DCC contains proteins such as MSL-1, MSL-2, MSL-3, MOF, MLE, and CLAMP, and two ncRNAs, *roX1* and *roX2*. We examined the FlyAtlas2 fly tissue expression data [27], and found that the ncRNAs *roX1* and *roX2* are depleted several fold in testis compared to most other tissues, while transcripts of the protein components of this complex are expressed more uniformly (S12 Fig).

In our scRNA-seq data, we found low-level expression of transcripts from all protein components of the DCC in early germ cells but found them less expressed in somatic cells (Fig 4A). Compared to other cell types, in GSC/early spermatogonia the only germline enrichments of genes involved in dosage compensation were *Clamp* and *msl-3* (adjusted p values 1.38E-09 and 5.41E-15, respectively). Conversely, we detected robust *roX1* enrichment in somatic hub and cyst cells (adjusted p values 1.52E-23, 0.00E00, respectively), and *roX2* enrichment in cyst cells (adjusted p value 1.12E-186), but germline expression was undetectable, concordant with the *roX* gene depletion observed by FlyAtlas2 (S12 Fig).

To confirm these findings, we performed RNA fluorescent in situ hybridization (RNA-FISH) on *roX1*, *roX2*, *msl-1*, *msl-2*, *msl-3*, and *mle* (Fig 4B-4G). As dosage compensation involves many peripheral and downstream proteins, we focused our RNA-FISH work on the core components of the MSL complex. RNA-seq and scRNA-seq data suggest that the expression levels of the DCC genes differ at the tissue and cell-type level, consistent with the RNA-FISH results. For example, *msl-3* is the most highly detectable component of the DCC in GSC and spermatogonia with RNA-FISH and scRNA-seq. *Mle* is detectable in the early germline in scRNA-seq, but we could not detect early germline *mle* expression with FISH. In FISH, we saw *mle* expression in peripheral muscle and pigment cells, concordant with its canonical role in somatic dosage compensation. The whole-tissue RNA depletion of *roX1* and *roX2* coincides with low/sparse expression in our testis scRNA-seq data as well as FISH images. FISH did not reveal significant pre-meiotic germline expression of any *msl* gene except msl-3. Many DCC genes such as *roX2*, *msl-1*, *msl-2*, and *msl-3* showed distinct nuclear localization in somatic muscle and pigment cells, which were not captured by scRNA-seq. Our RNA-FISH results are similar to those of earlier work, which found that of all the MSL proteins, only MLE showed germline protein expression and did not localize to the X chromosome [20].

To verify that the apparent absence of DCC transcripts in the germline was biological, not technical, we also performed RNA-FISH on *Drosophila* accessory glands with the same probes. All of the *msl* genes, as well as *roX1*, *roX2*, and *mle*, show distinct nuclear foci in accessory gland RNA-FISH (S13 Fig).

## Discussion

In this work, we directly observe dosage compensation in terms of X:autosome count ratios and differential gene activity around CES. These lines of evidence point to adult pre-meiotic dosage compensation for the X chromosome, consistent with recent findings in larval testis [9]. After meiosis, the X chromosome is downregulated to an extent that would be expected

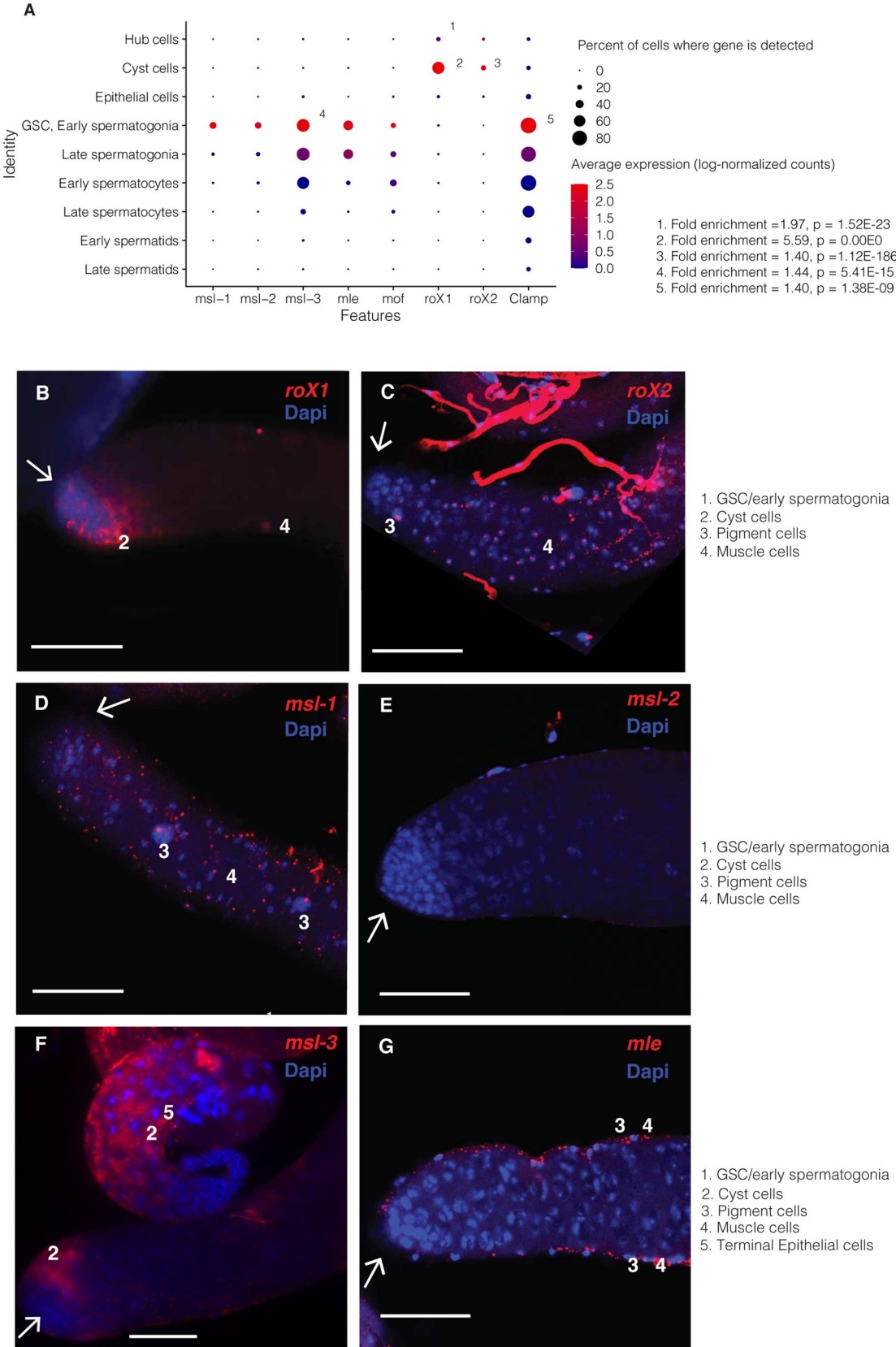

**Fig 4. Expression patterns of DCC components in early germ cells.** A) Expression patterns for MSL complex genes in testis scRNA-seq data, as well as *Clamp*, whose product binds the MSL complex to DNA. Dots are sized according to the percent of cells of a type where a transcript is detected and colored according to expression (log-normalized counts) of that gene in the cell type. Numbers indicate that a gene is enriched in that cell type compared to all other cells. P values are adjusted with Bonferroni's correction. *roX1* and *roX2* are enriched in cyst cells, *roX1* is enriched in hub cells, and *msl-3* is enriched in GSC/early spermatogonia. While we have found some expression of *msl* genes, most are not likely translated at high levels, as shown in [20]. B-G) RNA-FISH of the distal testis region. White arrows indicate the hub region. Scale bars are 50 μM for all images. B) *roX1* is usually but not always found in somatic cyst cells near the testes tip. In some images it is expressed in elongating spermatid cysts and in the nuclei of the somatic terminal epithelial cells. No nuclear expression was detected in germline cells at the distal tip (arrow). ScRNA-seq indicates expression in somatic cyst cells, but no significant germline expression. C) in FISH, *roX2* shows nuclear localization in peripheral muscle and pigment cells, with cytoplasmic expression detected in elongating spermatid cysts. Little to no expression was observed in the germ cell region. In scRNA-seq it shows 0 detectable counts in most cells of every type, but detectable counts more often in somatic cells than germ cells. D) in FISH, *msl-1 also* shows some expression in muscle and pigment cells at the testis periphery, but no such foci inside the testis. In scRNA-seq, by contrast, transcripts are faintly detectable in GSC/early spermatogonia. E) in FISH, *msl-2* was undetectable, but at higher exposure showed faint cytoplasmic expression throughout the region, and scRNA-seq finds weak, stochastic expression in early germ cells. F) in FISH, *msl-3* shows weak cytoplasmic expression in early cyst cells and elongating spermatids, as well as in the nuclei of muscle and pigment cells. In scRNA-seq data it is stochastically expressed in somatic cells but commonly expressed in early germ cells. G) in FISH *mle* shows bright foci in muscle and pigment cells but only weak expression in early stem cells (image is focused on peripheral cells at the testes edges and testis lumen). In scRNA-seq, expression levels peak in GSC/early spermatogonia and decline thereafter. This result is comparable to that of in [20], showing that *Mle* is expressed in germ cells but does not localize to the X chromosome.

from incomplete dosage compensation. Compared to prior methods which use transgenic reporters to query expression from various autosomal and sex chromosome loci [17,21], our approach provides a direct, holistic and high-throughput picture of whole-chromosome transcriptional trends in each cell type.

We found evidence for dosage compensation in three types of somatic cells and two types of pre-meiotic germ cells. These cells, however, express a different repertoire of DCC genes. Some somatic cells express *roX1* or *roX2*, but show sparse quantities of the *msl* genes and *mle*. In contrast, pre-meiotic germ cells show some expression of *msl-3* but *roX1* and *roX2* are barely detectable. This is concordant with depletion of *roX*, *msl-1*, and *msl-2* RNA in testis compared to other tissues in FlyAtlas2 RNA-seq data. However, RNA levels of the DCC genes do not correspond to the actual magnitude of dosage compensation in the adult male germline, suggesting that abundances of proteins and RNAs in the cells may not be well-correlated. Enriched gene expression around MSL CES sites in these cell types strongly implies some sort of active dosage compensation in somatic and early germ cells. This would normally indicate the activity of the DCC, but we were unable to detect significant germline expression of several components of the DCC with scRNA-seq or RNA-FISH. While the DCC proteins could be present and active without active transcription, the lack of germline *roX* expression is a more reliable clue for an alternative mechanism, since these are ncRNAs. In our scRNA-seq data, no DCC genes were enriched in germ cells except *msl-3* and *Clamp*, which were enriched in GSC/ early spermatogonia (Fig 4A). Rather than acting on dosage compensation, *msl-3* might instead be facilitating entry into meiosis, as suggested by recent work in mice [28]. If so, this could suggest a conserved function of *msl-3* across kingdoms and explain the lack of germline enrichment of any other DCC genes in our data.

While it is possible that a cell could perform dosage compensation without active transcription of DCC proteins, we would expect to find evidence of germline nuclear localization of *roX1* and *roX2* ncRNAs as we see in accessory glands. With RNA-FISH and scRNA-seq, we found evidence of germline *msl-3* expression, however, most of the other DCC components were so lowly expressed in the germline that we could only detect a few counts per cell with sequencing, and no enrichment with RNA-FISH. Our results do not rule out possible low levels of DCC proteins being present in pre-meiotic cells, but strongly demonstrate the absence of *roX1* and *roX2* in these cells. RNA-FISH suggests that the outer somatic layers of the testis

express most DCC genes, so their apparent depletion in scRNA-seq may be due to underrepresentation of the somatic pigment and muscle cells of the testis sheath.

Our results suggest that germline dosage compensation might occur without all of the canonical components of the DCC, consistent with earlier studies [16,20,29]. There could also be germline-specific unannotated isoforms of the *msl* genes that evade detection with RNA probes, gene annotations or antibodies tailored for variants discovered in somatic cells. Alternatively, the permissive chromatin environment of the testis [30] may be more conducive to dosage compensation, requiring only small amounts of *roX1* and *roX2* to boost X transcription to autosomal levels, levels that evade detection with RNA-FISH. On the other hand, other researchers [20] reported that H4K16ac is not enriched on the X in testes germ cells and the MSL proteins are undetectable in germ cells with immunostaining. This could indicate that the dosage compensation we observe may occur independently of the canonical DCC.

Since we observed likely germline activity of MSL CE sites, it is possible that there exists a noncanonical mechanism of early germline dosage compensation that uses the same sequence elements as somatic cells. This appearance of X chromosome "excess" dosage compensation is striking in GSC/early spermatogonia, and testis-biased and testis-specific genes seem disproportionately upregulated in GSC/early spermatogonia (S8, S10 and S11 Figs). The fact that these cell types still appear dosage compensated without these genes suggests that dosage compensation in these cell types is a mostly global phenomenon with uneven effects. Active CE sites in these cells may be connected to the appearance of excess dosage compensation in early germ cells, but this phenomenon may be linked to more cell-type specific upregulation of many X chromosome genes in early spermatogenesis. The excess dosage compensation in GSC/early spermatogonia may also be partly caused by upregulation of X-linked male-specific genes (S11A Fig, GSC/early spermatogonia p = 0.015, p.adj = 0.059) but not so in male-biased genes (S11B Fig, GSC/early spermatogonia $p < 0.001$). However, our current evidence does not support that this excess dosage compensation is completely independent of dosage compensation related mechanism as we still see excess dosage compensation after removing X-linked male-biased genes. The appearance of active CES in late spermatogonia indicates that dosage compensation (or a transcriptional trend with similar effects) is active in these cells and equalized X-autosome levels may not be solely due to retained transcripts produced from earlier cell stages.

Germline dosage compensation may perhaps be mediated by CLAMP, which has been shown to bind to MSL CE sites and increase X chromosome accessibility [31], and which we observed upregulated in early germ cells (Fig 4A). There is also a possibility that MSL recruitment is associated with other factors such as DNA replication [32], thus the MSL recruitment and the timing of dosage compensation in cells can be very dynamic.

Our results do not provide evidence to support or disprove MSCI in late germ cells. We observed, in spermatocytes and spermatids, that X chromosome genes are expressed with median RNA counts of 63–77 percent that of autosomal genes. This would be expected from incomplete or absent dosage compensation, as some transcripts present in these cells would be retained from earlier stages despite reduced X transcription. For us to confidently predict the presence of MSCI, we would have to observe X:autosome ratios $<0.5$ in meiotic and postmeiotic germ cells. While X:autosome ratios rise from 0.63 in late spermatocytes to 0.77 in late spermatids, this could be driven by reduced transcription of both the X and autosomes making leftover transcripts proportionally more abundant, or could be a technical artifact driven by globally lowered gene expression in late spermatids [33]. Alternatively, this phenomenon could arise due to increased chromatin accessibility of early germ cells, which might disproportionately affect the normally largely heterochromatic X chromosome.

While many exciting mechanistic questions remain, our study strongly supports the presence of dosage compensation in *Drosophila* pre-meiotic germ cells and shows that it occurs

despite sparse germline expression of *roX1*, *roX2* and most DCC proteins. Pre-meiotic dosage compensation appears to be driven from the same chromatin elements as somatic dosage compensation. Future studies should confirm that our observed trends of dosage compensation in scRNA-seq are abrogated when MSL CE sites are blocked, or that they occur during a testis-specific knockdown of *roX1* and *roX2*.

## Methods

### Integration and normalization of *D. melanogaster* scRNA-seq datasets

Raw reads from the two datasets were aligned to the FlyBase dmel_r6.15 reference GTF and Fasta [34] with Cellranger. The count matrices were made into separate Seurat objects, normalized and scaled with default Seurat parameters, and integrated with Seurat SCtransform. Neighbor finding, clustering and dimensional reduction were performed on the combined dataset. The exact parameters and commands used are available in the accompanying GitHub repository (see Data Availability). For all analyses, genes are only considered if they are detected in at least 3 cells, and cells are only considered if they express >200 genes. A gene is considered "expressed" in a cell if it has at least 1 read in that cell. Expression for Fig 4A was calculated with the NormalizeData function of Seurat, with a scale factor of 10000.

### Assigning cell types from the integrated scRNA-seq dataset

Cell types were assigned mostly the same marker genes from our previous work [23]. Clusters expressing *bam* and *aub* were assigned as a mixture of germline stem cells [35] and early spermatogonia [36]. Clusters with slight enrichment of *bam*, but less *His2Av* than GSC/early spermatogonia were interpreted as late spermatogonia. This was corroborated by the fact that this group is adjacent to the GSC/early spermatogonia cluster in the t-SNE. As we previously found that *fzo* expression peaks earlier than *twe* in spermatocytes, we assigned clusters with the highest enrichment of *fzo* [37] as early spermatocytes, and cells with *twe* [38] but no *fzo* as late spermatocytes [23]. We defined early spermatids as clusters with enriched *Dpy-30L2* but no *fzo* or *twe*, and clusters enriched for *p-cup* as late spermatids [39]. Cells enriched for *Fas3* were deemed somatic hub cells, clusters enriched for *zfh-1* were assigned as somatic cyst cells [40], and cells enriched for *MtnA* [41] but not *Fas3* were labelled somatic epithelial cells. One major update from our 2019 paper is that we used *Rab11*[25], not zfh1, as a marker gene for cyst cells, improving the cell-type assignments.

### Calculating relative RNA content from each cell type

RNA content per cell is the sum of all Unique Molecular Indices (UMIs/counts) detected from the X chromosome or autosomes in a cell type, divided by the number of cells. This is a proxy for the relative RNA content per cell and is not a measurement of the actual number of RNA molecules present.

### Comparing RNA output by chromosome and spermatogenic stage

For this method, we obtained the total RNA counts from every X and autosomal gene in every cell type. We then log transformed these counts with $y = Log2(counts+1)$ and performed non-directional Wilcoxon tests, with Holm-corrected p values indicating if genes from the X chromosome are likely to have equal median counts to genes from the autosomes.

## Correlating chromatin entry sites with nearby gene output

We obtained a list of MSL recognition sites [13] and converted the coordinates to *D. melanogaster* version 6 with the FlyBase coordinates converter [34](S3 Table). For every X chromosome gene, we then calculated the distance between the start coordinate of its gene region and the start coordinate of the closest chromatin entry sites. For each gene, we summed all the reads detected in each cell type. We log transformed these counts for every cell type with the equation Log2(counts+1) and calculated Pearson's R and corresponding p values between distance and gene counts for every cell type. We then adjusted these p values with Holm's method.

## RNA-FISH of DCC transcripts in *Drosophila* testis

Plasmids from Drosophila Gene Collection (DGC) libraries were used to generate Rox1 (CR32777), Mle (CG11680), MSL-1 (CG10385), MSL-2 (GH22488), MSL-3 (CG8631) DIG-labeled probes. The following primers were designed for Rox2 (CR32665) template production by PCR of genomic DNA: forward:

5'-AATTAACCCTCACTAAAGGGTTGCCATCGAAAGGGTAAATTG-3', reverse: 5'- GTAATACGACTCACTATAGGGCAGTTTGCATTGCGACTTGT-3'. RNA probe preparation and use were carried out as previously described [42,43]. Images were acquired using a Leica Sp8 confocal microscope.

## Calculating transcript enrichment in scRNA-seq data

For the dosage compensation genes, we used the FindAllMarkers Seurat function to identify genes enriched in a cell type compared to all other cells. We adjusted P values with Bonferroni's correction.

## Reproducibility

To ensure that our findings are robust with respect to biological and technical variability, we performed analyses from the main manuscript separately on each strain. The main findings from the paper are highly similar in both strains, with only small differences that do not negate our main findings. These results are in S3, S4, S5 and S7 Figs.

## Supporting information

**S1 Fig. Integrated t-SNE of both strains corresponding to Fig 1A.** Both strains overlap well. Cell types were assigned from this integrated dataset.
(PDF)

**S2 Fig. Relative enrichment of marker genes used to assign cell types.** These genes were used to assign cell types, with details in the methods section.
(PDF)

**S3 Fig. Marker genes used to assign cell types, split by strain.** This corresponds to S2 Fig. Marker enrichment in both strains corroborates cell type assignments.
(PDF)

**S4 Fig. RNA per cell from X and autosomes, for each of the two strains separately.** Left is R517, right is wild type. This corresponds to Fig 1C and 1D.
(PDF)

**S5 Fig. Median counts from X and autosome, split by strain.** Corresponding to Fig 2, both strains support the appearance of pre-meiotic dosage compensation, somatic dosage compensation, and meiotic and post-meiotic X downregulation. In addition, in both strains, GSC and early spermatogonia appear to show X over-compensation. P values are from a two-tailed Wilcoxon test of the null hypothesis that X and autosomal counts are equal, adjusted with Holm's correction.
(PDF)

**S6 Fig. Scaled expression provides evidence of cell-type-biased dosage compensation.** A) Boxplots indicate the distribution of scaled expression of autosomal and X chromosome genes within each cell type. 0 represents a gene's mean expression across all cell types. In hub cells, epithelial cells, and premeiotic germ cells, scaled expression of X genes exceeds that of autosomal genes, suggesting that these cells experience X chromosome dosage compensation. Asterisks represent Holm-adjusted p values of directional Wilcoxon tests. B) Scaled expression of Y chromosome genes exceeds that of the autosomes in late spermatocytes, early spermatids, and late spermatids. This indicates that after meiosis, Y chromosome genes are not downregulated to the same extent as autosomal genes. Asterisks represent p values as follows: ns: $>0.05$, $^*<0.05$, $^{**}<0.005$, $^{***}<0.0005$, $^{****}<0.00005$.
(PDF)

**S7 Fig. Relationship between CES proximity and gene expression, grouped by cell type and strain.** Both datasets agree that somatic and pre-meiotic cells have a statistical enrichment of counts detected from genes within 10000 bp from a CES. In the wild dataset, however, proximal genes are enriched in every cell type, although less so in meiotic and post-meiotic cells than in pre-meiotic and somatic cells.
(PDF)

**S8 Fig. Evidence that Fig 3 is not confounded by testis-specific or testis-biased genes.** Shown is the analysis from Fig 3, repeated with testis-specific or testis-biased genes removed from the analysis. Neither result changes the conclusions in Fig 3.
(PDF)

**S9 Fig. Close chromatin entry sites correlate with increased transcription of X chromosome genes in cell types experiencing DC.** A) Each dot is an X chromosome gene; the X axis is the distance (in bp) between the gene start and the closest Chromatin Entry Site (CES) from Alekseyenko et al. 2008. The Y axis is the log-transformed sum of all counts of that gene in a cell type for every gene. The black line is a Loess regression showing an approximate trend between the two axes. B) Pearson's R shows that CES distance loosely correlates with RNA counts in hub, cyst, epithelial, GSC, early spermatogonia, and late spermatogonia cells, all cell types where we found evidence of dosage compensation. Spermatocytes and spermatids have Pearson's R closer to zero than DC-exhibiting cells, indicating less of a relationship between CES distance and transcription. In addition, these non-dosage-compensated cell types have a high Holm-adjusted p value, suggesting that distance and counts are not likely correlated in these cells.
(PDF)

**S10 Fig. Evidence that Fig 1 is not confounded by testis-specific or testis-biased genes.** Each panel corresponds to Fig 1B, with testis-specific or testis-biased genes removed from the dataset. Overall patterns of RNA per cell do not significantly change.
(PDF)

**S11 Fig. Testis-specific and testis-biased genes do not confound the inference of pre-meiotic dosage compensation in Fig 2.** This is the same analysis as Fig 2, with testis-specific genes removed. The overall patterns of pre-meiotic dosage compensation and meiotic X downregulation are preserved, indicating that these genes did not influence our main findings. One difference from Fig 2 is that when testis-specific genes are removed, there is no longer statistical enrichment of X chromosome genes in GSC and early spermatogonia, indicating that testis-specific genes contribute to the appearance of X-chromosome overcompensation in these cells. Another difference is that without testis-biased genes, late spermatogonia no longer show statistical depletion of X chromosome counts, suggesting that these genes contribute to the earliest signs of reduced dosage compensation.
(PDF)

**S12 Fig. DCC gene enrichment in FlyAtlas2 data.** For each gene in the DCC, we queried the FlyAtlas2 website for their calculated enrichments in selected tissues compared to all tissues. *RoX1*, *roX2*, *msl-1* and *msl-2* are very depleted in testis compared to somatic tissues, corresponding to their relatively stochastic expression in our scRNA-seq and RNA-FISH data.
(PDF)

**S13 Fig. Additional DCC gene RNA-FISH images.** A) *roX1* antisense negative control in testis. B) *roX1* antisense negative control in the accessory gland. C) *roX1* shows nuclear expression in accessory gland main cells and diffuse puncta in accessory gland lumen, as well as membrane expression in secondary cells. D) In the accessory gland, small patches of *roX2* are expressed in the cytoplasm and nuclei of main cells and along membranes of secondary cells. *roX2* is prevalent in accessory gland lumen as diffuse foci. E) *mle* is expressed as dense foci localized in nuclei of main cells, with smaller foci in the cytoplasm of main cells. In secondary cells, it is enriched near membranes and shows smaller and fewer dots in the accessory gland lumen. F) *msl-2* shows small foci distributed at low levels in main cell cytoplasm. In secondary cells, it is enriched along membranes and in discrete foci in the accessory gland lumen. G) *msl-3* shows strong expression patterns in the cytoplasm of main cells and the membranes of secondary cells, with smaller diffuse foci distributed in the accessory gland lumen.
(PDF)

**S1 Table. Mean counts per cell from the X chromosome and autosomes in every cell type, corresponding to Fig 1B.** The X chromosome is most highly transcribed compared to autosomes in GSC and spermatogonia.
(DOCX)

**S2 Table. Correlation between distance and transcription for X chromosome genes by cell type.** Corresponding to S9 Fig, this is the Pearson's R between distance and log2(counts+1) for every gene per cell type, with Holm-adjusted p values. Cell types with evidence of dosage compensation are shown with an asterisk.
(DOCX)

**S3 Table. Chromatin entry sites from Alekseyenko et. al 2008 [13].** We have converted the start and end coordinates to FlyBase R6 version using the FlyBase coordinates converter.
(DOCX)

## Acknowledgments

We thank the Zhao lab members for useful discussion and critical reading of the manuscript, especially Nicolas Svetec for providing essential ideas.

## Author Contributions

**Conceptualization:** Evan Witt, Li Zhao.

**Data curation:** Evan Witt, Zhantao Shao, Chun Hu, Henry M. Krause, Li Zhao.

**Formal analysis:** Evan Witt, Zhantao Shao, Chun Hu, Henry M. Krause.

**Funding acquisition:** Henry M. Krause, Li Zhao.

**Investigation:** Evan Witt, Zhantao Shao, Chun Hu, Henry M. Krause, Li Zhao.

**Methodology:** Evan Witt, Zhantao Shao, Chun Hu, Henry M. Krause, Li Zhao.

**Resources:** Evan Witt, Zhantao Shao, Chun Hu.

**Software:** Evan Witt.

**Supervision:** Henry M. Krause, Li Zhao.

**Validation:** Evan Witt, Zhantao Shao, Chun Hu, Henry M. Krause.

**Visualization:** Evan Witt, Zhantao Shao, Chun Hu, Henry M. Krause, Li Zhao.

**Writing – original draft:** Evan Witt, Li Zhao.

**Writing – review & editing:** Evan Witt, Li Zhao.

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
