## [Decision Letter · Decision Letter 0]

4 Jun 2021

Dear Dr Zhao,

Thank you very much for submitting your Research Article entitled 'Single-cell RNA-sequencing reveals pre-meiotic X-chromosome dosage compensation in Drosophila testis' to PLOS Genetics.

The manuscript was fully evaluated at the editorial level and by independent peer reviewers. The reviewers appreciated the attention to an important problem, but raised some substantial concerns about the current manuscript. Based on the reviews, we will not be able to accept this version of the manuscript, but we would be willing to review a much-revised version. We cannot, of course, promise publication at that time.

Should you decide to revise the manuscript for further consideration here, your revisions should address the specific points made by each reviewer. We will also require a detailed list of your responses to the review comments and a description of the changes you have made in the manuscript. There are two major issues that are particularly important to address:

1. The current manuscript’s use of “excess dosage compensation” should be clarified. Reviewer 3 states that the conclusion of overcompensation is likely due to the inclusion of X-linked testes-specific genes. Reviewer 2 also points out that, if the appearance of excess dosage compensation is due to gene-specific regulation, then it is not actually a form of dosage compensation at all. The claim of “excess dosage compensation” should be re-evaluated in light of the reviewer’s comments. If the excess transcription on the X chromosome relative to autosomes in GSC/early spermatogonia is due to high expression of testes-specific genes, it is not accurate to refer to this phenomenon as “excess dosage compensation” and the manuscript should be revised accordingly. On the other hand, if gene-specific regulation does not explain the excess of transcription from the X chromosome, then the excess dosage phenomenon should be discussed further, as suggested by Reviewer 2 (points 9 & 12).

2. *clamp* should be added to Figure 4A and a justification of why *clamp* was not used for RNA FISH should be included in the manuscript text. The RNA FISH results should also be explicitly compared to the immunostaining results from Rastelli & Kuroda (1998), as suggested by Reviewer 1.

If you decide to revise the manuscript for further consideration at PLOS Genetics, please aim to resubmit within the next 60 days, unless it will take extra time to address the concerns of the reviewers, in which case we would appreciate an expected resubmission date by email to plosgenetics@plos.org.

[LINK]

We are sorry that we cannot be more positive about your manuscript at this stage. Please do not hesitate to contact us if you have any concerns or questions.

Yours sincerely,

Christopher E Ellison

Guest Editor

PLOS Genetics

Gregory P. Copenhaver

Editor-in-Chief

PLOS Genetics

Reviewer's Responses to Questions

**Comments to the Authors:**

Reviewer #1: Attachment uploaded.

Reviewer #2: The authors used scRNA-seq data from one of their previous publications to investigate dosage compensation (DC) in testis of adult D. melanogaster. They found evidence for complete DC in somatic and pre-meiotic cells, where X:autosome expression ratios were close to 1. The ratio decreased in germline stem cells and early spermatogonia, and further decreased after meiosis. They conclude that there is some level of pre-meiotic DC that occurs independent of roX RNAs and the canonical DCC.

Some of the results presented in this manuscript have been reported previously (complete DC in soma, lack of DC in testis), but have been somewhat controversial. By using scRNA-seq, the authors are able to address these issues at a finer scale than previous studies that focused on whole tissues or dissected regions of tissues. Thus, I think they have generated valuable data that extend our knowledge of this topic. The data are available in a format convenient for researchers and should be an important resource. There is, however, a need for the authors to improve their presentation and expand upon some of their analyses.

1. line 21: "X chromosome transcription is equalized in the somatic cells of both males and females" - not phrased clearly. Does DC equalise expression of X between sexes? Or between X and autosomes within males? The latter is what the authors investigate in the manuscript.

2. line 40: should be "needs"

3. line 51: authors don't define some abbreviations (DCC, MSL, MLE, MOF). Readers will probably be confused.

4. line 68: "Other work suggests that demasculinization of the X chromosome might be partly due to dosage compensation in Drosophila (Bachtrog et al., 2010)." - I think more explanation is needed here. Why would DC lead to demasculinization? One might expect the *absence* of DC to lead to demasculinisation.

5. lines 70-83: there are some awkward references to previous literature in which the researchers are mentioned by name apart from the citations, such as "Meiklejohn and Presgraves", "Rastelli & Kuroda", "Parsch group". It would be better to only cite the papers.

6. line 75: "the mechanism of hypothetical germline DC is an additional mystery." How can there be a mechanism of something that is hypothetical?

7. line 82: "multiple transgenic insertions in X and autosomes made by the Parsch group (Hense et al., 2007; Kemkemer et al., 2011) show that X inactivation exceeds that expected for loss of dosage compensation" - this is true, but these transgene experiments also controlled reporter gene dose to always be one (whether on X or autosome), so gene dose or loss of DC can be ruled out as a cause for an expression difference. The same approach was used by Landeen et al. (PLoS Biol. 2016 Jul 12;14(7):e1002499).

8. Fig 1B: it might help to indicate which gene you used as a marker for each cell type in parentheses next to the names of the cell types on the figure.

9. Table 1: there seems to be a transition from "excess DC" in early spermatogonia to "DC" in late spermatogonia to "no DC" in early spermatocytes. Since this is a developmental progression, I am not sure about the conclusion in early spermatogonia. This is because the authors measure RNA content, but do not measure transcriptional activity directly. If there is an excess of X RNA in an early stage, then even if there is no DC in the next stage one might expect to see more X RNA simply because it remains from the previous stage and has not yet degraded. To me, the data seem to be in agreement with this interpretation. The unusual observation is that there is excess DC in GSC and early spermatogonia. After these stages, the results could be consistent with "no DC".

10. line 196: "close to an MSL CES" - MSL and CES sites are not always the same. I think you only need "CES" here. Regarding the effect of DCC distance on X expression in somatic or germline cells, a relevant reference is Belyi et al. (Genome Biol Evol. 2020 Dec 6;12(12):2391-2402).

11. line 337" "MSL CE sites" - see above comment. Should be CES?

12. Discussion: As mentioned above, one of the most striking findings of the study is that there is "excess DC" in GSC/early spermatogonia. The authors should provide further discussion of this. Is the effect chromosome-wide? Could it be a result of gene-specific regulation and, thus, not a form of dosage compensation?

13. line 374: "Witt et. al 2019." - put parentheses around the year. The period should be after "al."

14. line 344, Discussion of MSCI: was there a relationship between the expression level of genes and the X:autosome ratio? Previous work suggests that if one considers only genes with very high expression the X:autosome is reduced more than if genes with lower expression are included (see Argyridou et al. (Genes (Basel). 2018 May 4;9(5):242.)

.

Reviewer #3: The authors used single cell RNA-sequencing data to analyze dosage compensation (DC) in Drosophila testis, whose presence has been debated previously. By analyzing Unique Molecular Indices (UMI, Fig. 1) across X and autosomes, read count (Fig. 2) and expression correlation against Distance to closest Chromatin Entry Site (CES) (Fig. 3), authors argued the presence of DC in spermatogonia (pre-meiosis cells) and somatic cells in testis. Then, authors examined the expression of four key protein-coding genes (e.g. MLE, MSL-1) of dosage compensation complex (DCC) and two key noncoding RNAs (roX1, roX2). Authors additionally analyzed testis-specific genes and believed that these genes are overcompensated.

The concept of DC is important. Authors revealed several lines of evidence supporting the presence of DC in Drosophila testis and potentially novel mechanistic insights on the machinery of DC (likely non-canonical). Given these two lines of consideration, I think that this manuscript could be publishable. However, the current version has a room to be significantly improved.

Major concerns:

1. The strong expression of X-linked testis-specific genes have been interpreted as overcompensation. However, the consequence of DC is to balance the expressional output between X and autosome. So, for these testis-specific genes, DC seems not necessary. This is why testis-specific genes have been generally excluded in the analysis of expression ratio between X and autosome (X:A ratio, e.g. Pubmed ID: 28132849).

2. The writing or figure design is problematic.

a) Fig2/Table1 and Fig3/Table2 have been designed in the same way where X and autosome have been shown side by side, which is followed by statistics in Tables. I wonder why Fig1 (UMI distribution) is not designed in the same way.

b) To highlight the key discovery of this work, I suggest authors to add a figure in Discussion to summarize the dynamic picture of DC, the underlying complexity of DC machinery (non-canonical) and how this work is in line with the previous related work (e.g. Mahadevaiah et al., 2020; Mahadevaraju et al., 2020). In this aspect, authors mentioned “msl-3 might instead be facilitating entry into meiosis…the lack of germline enrichment of any other DCC genes”. Does this mean that authors believed the machinery underlying pre-meiotic DC is entirely different from the canonical one?

3. Authors mentioned Clamp as one essential protein of DCC. However, in fig. 4, this gene is not covered. Could authors explain why? In addition, “no DCC genes were enriched in germ cells except msl-3 and Clamp, which were enriched in GSC/early spermatogonia (Figure 4A).” Again, Clamp is not shown in Fig. 4A.

Minor concerns:

1. “others to suppress it in females”. I guessed that authors referred to human system. However, for human, an additional mechanism acts to upregulate single active X chromosome to balance X and autosomal transcription. So, writing should be revised here to be more specific.

**Have all data underlying the figures and results presented in the manuscript been provided?**

Reviewer #1: Yes

Reviewer #2: Yes

Reviewer #3: Yes

PLOS authors have the option to publish the peer review history of their article (what does this mean?). If published, this will include your full peer review and any attached files.

Reviewer #1: **Yes: **Leila E Rieder

Reviewer #2: No

Reviewer #3: No

---

## [Decision Letter · Decision Letter 1]

20 Jul 2021

Dear Dr Zhao,

We are pleased to inform you that your manuscript entitled "Single-cell RNA-sequencing reveals pre-meiotic X-chromosome dosage compensation in Drosophila testis" has been editorially accepted for publication in PLOS Genetics. Congratulations!

Please note that Reviewer #1 has a suggestion (see below) for clarifying a sentence in the abstract.  Feel free to address that as you prepare your final draft for the production team (the editorial team will not need to re-evaluate).

Yours sincerely,

Christopher E Ellison

Guest Editor

PLOS Genetics

Gregory P. Copenhaver

Editor-in-Chief

PLOS Genetics

Comments from the reviewers (if applicable):

Reviewer's Responses to Questions

**Comments to the Authors:**

Reviewer #1: I am satisfied with the changes made by the authors. I found the response thoughtful and careful.

One suggestion: The opening abstract sentence changed in response to reviewer 2 now reads: "Dosage compensation is a mechanism by which X chromosome transcription is equalized to that of autosomes in the somatic cells of both males and females." (line 23-24)

This is still confusing, as dosage compensation in flies only occurs in males (mammals only in females), and this sentence suggests that it occurs in both sexes. I believe that this is what Reviewer 2 meant in their original review. The authors should clarify. Something like:

"Dosage compensation equalizes X-linked expression between X males and XX females. In male fruit flies, expression levels of the X-chromosome are increased two-fold* to compensate for their single X chromosome."

*the fold increase is actually more like 1.4 fold (Hamada, Genes&Dev, 2005).

Reviewer #2: The authors have done a good job of responding to my concerns and have revised the manuscript appropriately. I have no further comments.

Reviewer #3: This version has been significantly improved and I do not have additional comments.

**Have all data underlying the figures and results presented in the manuscript been provided?**

Reviewer #1: Yes

Reviewer #2: Yes

Reviewer #3: Yes

PLOS authors have the option to publish the peer review history of their article (what does this mean?). If published, this will include your full peer review and any attached files.

Reviewer #1: **Yes: **Leila Rieder

Reviewer #2: No

Reviewer #3: No

**Data Deposition**

http://datadryad.org/submit?journalID=pgenetics&manu=PGENETICS-D-21-00554R1

**Press Queries**

---

## [Editor Report · Acceptance letter]

12 Aug 2021

PGENETICS-D-21-00554R1 

Single-cell RNA-sequencing reveals pre-meiotic X-chromosome dosage compensation in Drosophila testis  

Dear Dr Zhao, 

We are pleased to inform you that your manuscript entitled "Single-cell RNA-sequencing reveals pre-meiotic X-chromosome dosage compensation in Drosophila testis " has been formally accepted for publication in PLOS Genetics! Your manuscript is now with our production department and you will be notified of the publication date in due course.

With kind regards,

Andrea Szabo

PLOS Genetics

On behalf of:
